# A C21-steroidal derivative suppresses T-cell lymphoma in mice by inhibiting SIRT3 via SAP18-SIN3

Babu Gajendran [1,2], Krishnapriya M. Varier[3], Wuling Liu[1,2], Chunlin Wang[1,2], Klarke M. Sample[4], Eldad Zacksenhaus[5,6], Cui Juiwei[7], LieJun Huang[1,2 ✉], XiaoJiang Hao[1,2 ✉] & Yaacov Ben-David[1,2 ✉]

The SIN3 repressor complex and the NAD-dependent deacetylase SIRT3 control cell growth, and development as well as malignant transformation. Even then, a little known about cross-talks between these two chromatin modifiers or whether their interaction explored therapeutically. Here we describe the identification of a $C_{21}$-steroidal derivative compound, 3-O-chloroacetyl-gagamine, A671, which potently suppresses the growth of mouse and human T-cell lymphoma and erythroleukemia in vitro and preclinical models. A671 exerts its anti-neoplastic effects by direct interaction with Histone deacetylase complex subunit SAP18, a component of the SIN3 suppressor complex. This interaction stabilizes and activates SAP18, leading to transcriptional suppression of *SIRT3*, consequently to inhibition of proliferation and cell death. The resistance of cancer cells to A671 correlated with diminished SAP18 activation and sustained SIRT3 expression. These results uncover the SAP18-SIN3-SIRT3 axis that can be pharmacologically targeted by a $C_{21}$-steroidal agent to suppress T-cell lymphoma and other malignancies.

[1] State Key Laboratory of Functions and Applications of Medicinal Plants, Guizhou Medical University, Guiyang 550014, China. [2] The Key Laboratory of Chemistry for Natural Products of Guizhou Province and Chinese Academic of Sciences, Guiyang 550014, China. [3] Department of Medical Biochemistry, Dr. A.L.M. Post Graduate Institute of Basic Medical Sciences, University of Madras, Taramani Campus, Chennai 600113, India. [4] The National Health Commission's Key Laboratory of Immunological Pulmonary Disease, Guizhou Provincial People's Hospital, The Affiliated Hospital of Guizhou University, Guiyang, Guizhou 550002, China. [5] Department of Medicine, University of Toronto, Toronto M5G 2M1 ON, Canada. [6] Division of Advanced Diagnostics, Toronto General Research Institute-University Health Network, Toronto, ON, Canada. [7] The Cancer Center of the First Hospital of Jilin University, Changchun, Jilin 130021, China. ✉email: huangliejun@126.com; haoxj@mail.kib.ac.cn; yaacovbendavid@hotmail.com

T-cell lymphomas represent diverse malignancies, including T-cell lymphoma/leukemia (ATLL), mature T/NK cell lymphomas (MTNKL), cutaneous T-cell lymphomas (CTCL), and others[1–5]. Current conventional treatment for T-cell lymphomas involves combinations of chemotherapy, radiotherapy, and surgery[6–8]. In the last two decades, small molecules drugs developed for better treatment of this diverse type of malignancies. Histone deacetylase inhibitors (HDACi) have recently emerged as major targets for therapy[9]. The HDACi, Vorinostat, Istodax, and Belinostat, have been FDA-approved for the treatment of T-cell lymphomas[10–13]. Identification of alternative inhibitors for specific HDAC classes[14] may further improve therapy for this deadly disease.

The histone deacetylase complex SIN3 is a master scaffold that acts as a transcriptional repressor when bound to HDACs and other proteins[15]. SIN3 Associated Protein 18 (SAP18) is a subunit of SIN3 that enhances SIN3-mediated transcriptional repression through binding to SIN3 complex containing promoters[16]. SAP18 functions as a tumor suppressor and its loss represses invasion and angiogenesis of Kaposi's sarcomas[17]. NAD-dependent protein deacetylase sirtuin-3 (SIRT3), a class III HDAC[14], regulates metabolic processes in the mitochondria[18] and nucleus[19]. SIRT3 expression promotes tumorigenesis, including lymphomagenesis[20–22]. So far, a connection between SIN3/SAP18 and SIRT3 not been established, nor is it known if such interaction impacts cancer and whether it exploited therapeutically.

By screening a series of $C_{21}$-steroidal derivatives, we identified herein the compound 3-O-chloroacetyl-gagamine (A671), which exerts potent therapeutic activity against T-cell lymphoma and erythroleukemia. We show that A671 acts as an HDAC-class III inhibitor that suppresses transcription of SIRT3, leading to apoptosis in culture and inhibition of leukemia and lymphoma in vivo. Mechanistically, we show that A671 directly interacts with SAP18, inducing activation of the SAP18/SIN3 complex, and transcriptional silencing of SIRT3. These results uncover unknown regulation of SIRT3 by the SIN3 complex via SAP18, and identify A671 as a potent inhibitor for the treatment of erythroleukemia, T-cell lymphoma, and possibly other malignancies.

## Results

### A671 inhibits T-cell lymphoma and erythroleukemia in vitro.
A library of 72 $C_{21}$-steroidal derivatives, including 69 pregnane-type steroids we have previously reported[23] and additionally, newly gagamine derivatives A670–A672 (Supplementary Note), has been screened for growth inhibition of several cancer cell lines using MTT assay. Only one compound, A671, possessed growth inhibitory activity. The structure of A671 and two related compounds, A670 and A672, is depicted in Fig. 1a. A671 selectively suppressed cell growth and viability of T-cell lymphoma (EL4, MOLT4, CEM-C7H2, and BW-5147), erythroleukemia (HEL, K562, DP17-17, CB7, and CB3), and B-cell lymphoma (Raji, OCI_LY10, TK6) cells, but had only a mild or negligible activity against cell lines representing other cancer types including breast and prostate cancer as well as multiple myeloma (Fig. 1b). It strongly inhibited the growth of one melanoma cell line (WM9) but not another (WM239; Fig. 1b). These results indicate a selective anticancer activity of A671 with a preference for T-cell lymphomas and erythroleukemia. A671 reduced cell proliferation (Fig. 1c) and promoted apoptosis (Fig. 1d, e) of the T-cell lymphoma (EL4), and erythroleukemia (HEL) cell lines (Supplementary Fig. 1a, b). It also altered cell cycle progression leading to higher $G_1$ and lower S and $G_2/M$ phases in both cell lines (Fig. 1f and Supplementary Fig. 1c). Treatment of HEL cells with A671 resulted in a lower expression of the growth-promoting

phospho-MAPK kinase, phospho-AKT, and the anti-apoptotic protein BCL2, but not affected the expression of other factors such as c-MYC and FLI1 (Fig. 2a).

### A671 suppresses T-cell lymphoma and erythroleukemia in mice.
Next, we examined the effect of A671 in vivo using mouse models of T-cell lymphoma and erythroleukemia. Murine T-cell lymphoma derived EL4 cells ($5 \times 10^6$) intravenously injected into C56BL/6 mice. One week later, mice ($n = 6$/group) were treated every other day with vehicle alone (DMSO) or A671 (3 mg/kg) for 2 weeks. As shown in Fig. 2b, vehicle-treated mice succumbed to lymphoma ~70 days post-injection. In contrast, most A671-treated mice survived for more than 120 days post-injection ($P = 0.018$).

To model erythroleukemia, newborn BLAB/c mice ($n = 8$/group) infected with friend murine leukemia virus (F-MuLV). Leukemic cells were detectable as early as 3–5 weeks post-viral infection[24], whereas frank leukemia developed after 2–3 months through activation of Fli-1 by retroviral insertion[25–27]. Mice were treated every other day for 2 weeks with A671 (3 mg/kg), starting at five weeks post-infection. A671 treatment resulted in robust inhibition of leukemogenesis compared to control DMSO treated mice (Fig. 2c; $P = 0.0027$). While control leukemic mice become severely anemic as determine by hematocrit analysis, A671 administered group had higher values indicating lower erythropoiesis suppression (Fig. 2d). Hematocrit values in nonleukemic control mice were near 45–50%[24]. No difference in spleen weight observed at the time of death between A671 and control groups (Fig. 2e). Thus, A671 is a potent inhibitor of T-cell lymphoma and erythroleukemia both in culture and in preclinical models.

### A671 inhibits HDAC class III proteins.
HDACs classified into four groups based on sequence homology to the yeast enzymes and domain organization. Class I, II, and IV, considered "classical" HDACs, have a zinc-dependent active site and are sensitive to trichostatin A (TSA); class III represents a family of $NAD^+$-dependent enzymes known as sirtuins that are resistant to TSA but sensitive to nicotinamide[14,28,29]. Given the importance of histone modifications in T-cell lymphomas[9] and specificity of A671 to this type of malignancy (Fig. 1b), we examined these classes of HDACs as potential targets of A671. This analysis revealed that A671 had a marginal inhibitory effect on the activity of HDAC Class I, IIA, and IIB (Fig. 3a), whereas TSA inhibited these HDACs at nanomolar concentrations (Fig. 3b). In contrast, A671 effectively suppressed HDAC class III (Fig. 3c). Notably, while the positive control, nicotinamide, inhibited HDAC III class at millimolar concentrations, A671 exerted the same level of inhibition at the micromolar range (Fig. 3d vs. c). We concluded that A671 is a potent HDAC class III inhibitor.

### A671 binds to SAP18 and activates SIN3 repression.
Since A671 targets HDAC class III activity, we next determined its effect on the transcription of SIRT1–3, SIRT5, and SIRT7 (Fig. 3e–i), which predominantly expressed in hematopoietic cells[14]. Using Q-RT-PCR analysis, in EL4 cells, A671 significantly increased SIRT1 and SIRT2 mRNA levels but did not affect SIRT5, whereas SIRT3 and SIRT7 were moderately or strongly downregulated (Fig. 3g–i). These results are consistent with previous reports implicating SIRT3 and SIRT7 in cell survival and tumor progression[30–32].

Next sought to identify factors that regulate SIRT3 expression in response to A671. To this end, we performed RNA-sequencing analysis on HEL cells treated with A671 and searched for genes that show reverse correlation with SIRT3. Interestingly, SIRT3 downregulation (0.49-fold, $P = 0.032$) strongly correlated with

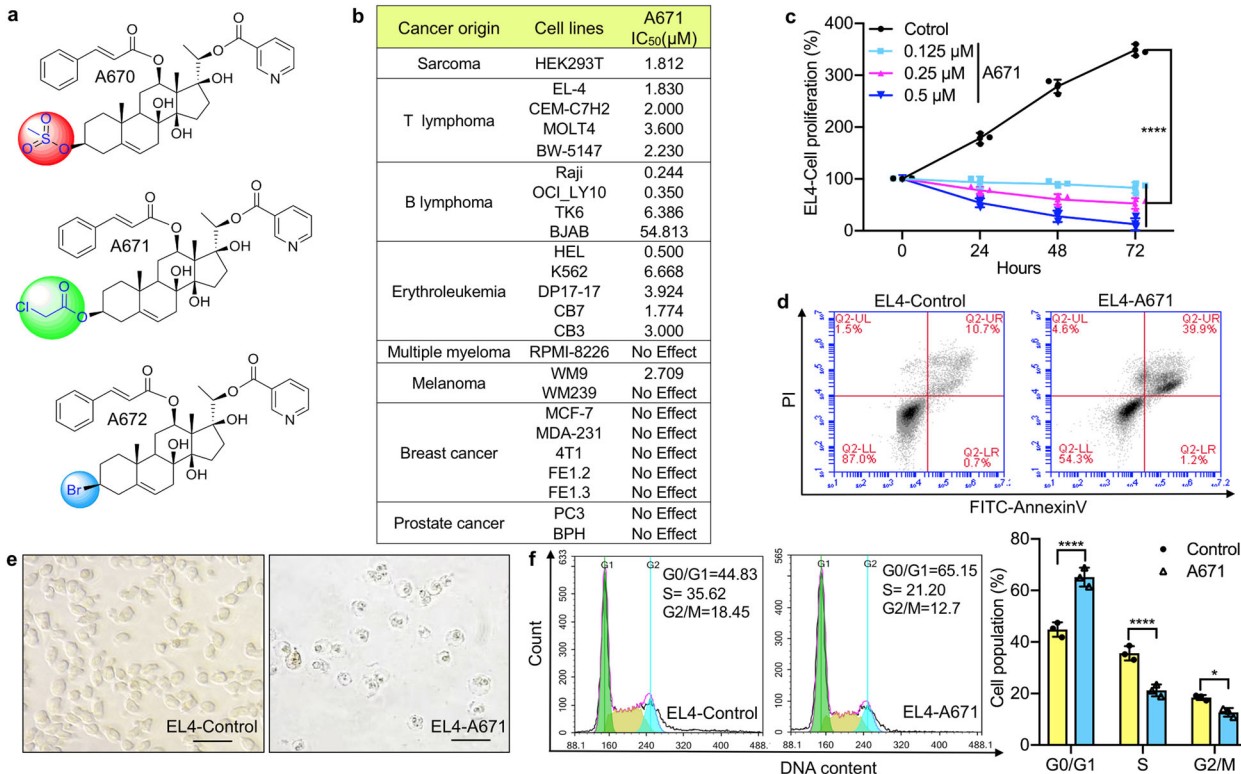

**Fig. 1 The C$_{21}$-steroidal derivative, A671, induces T-cell lymphoma cell death. a** Chemical structure of A670–A672. **b** IC$_{50}$ analysis for A671 in indicated cell lines. **c** Inhibition of T-cell lymphoma (EL4) cell proliferation by indicated doses of A671. **d** Induction of apoptosis in EL4 cells treated for 24 h with A671 (0.5 μM). **e** Image of apoptotic EL4 cells after 24 h treatment with A671 (0.5 μM; magnification ×100). **f** G$_2$/M cell cycle arrest after 24 h treatment of EL4 cells with A671 (0.5 μM). $P < 0.0001$ (****) by two-tailed student $t$-test.

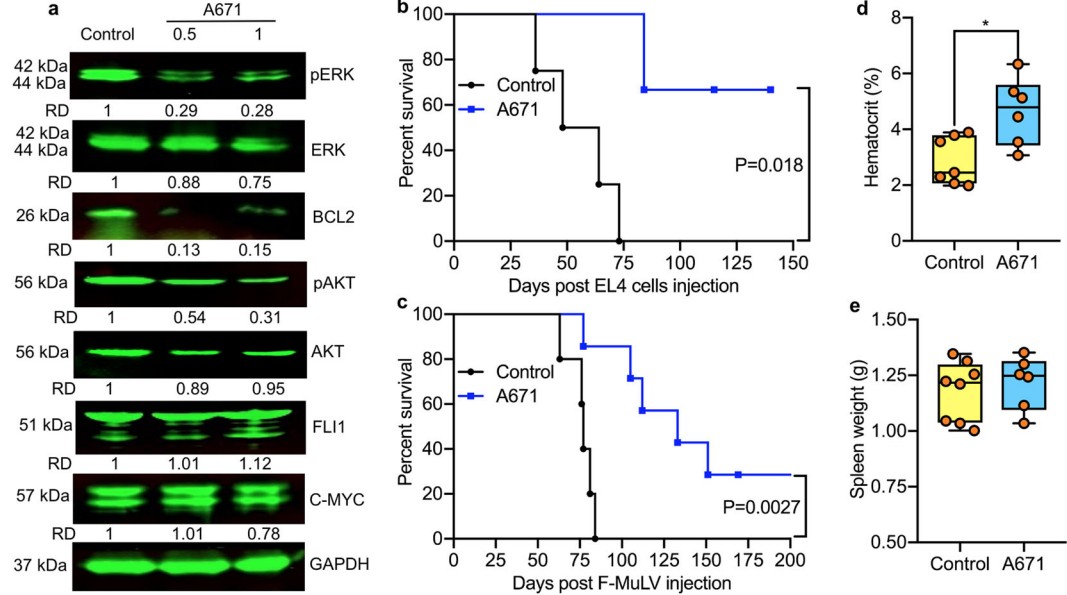

**Fig. 2 A671 inhibits the progression of T-cell lymphoma and erythroleukemia in preclinical models. a** HEL cells were treated for 12 h with A671 (0.5 μM), lysed, and cell extracts analyzed by Western blot for the indicated antibodies. RD relative density. **b**, **c** C57BL/6 mice with T-cell lymphoma induced following injection of EL4 cells (**b**) or with erythroleukemias induced 5 weeks after F-MuLV injection (**c**) were treated with A671 (3 mg/KG) every other day for 2 weeks. Regression analysis for tumor growth calculated by student $t$-test and death was used to plot a Kaplan–Meier survival curve. **d**, **e** Hematocrit (**d**) and tumor weight (**e**) of erythroleukemic mice at the onset of morbidity. $P < 0.05$ (*) by two-tailed student $t$-test.

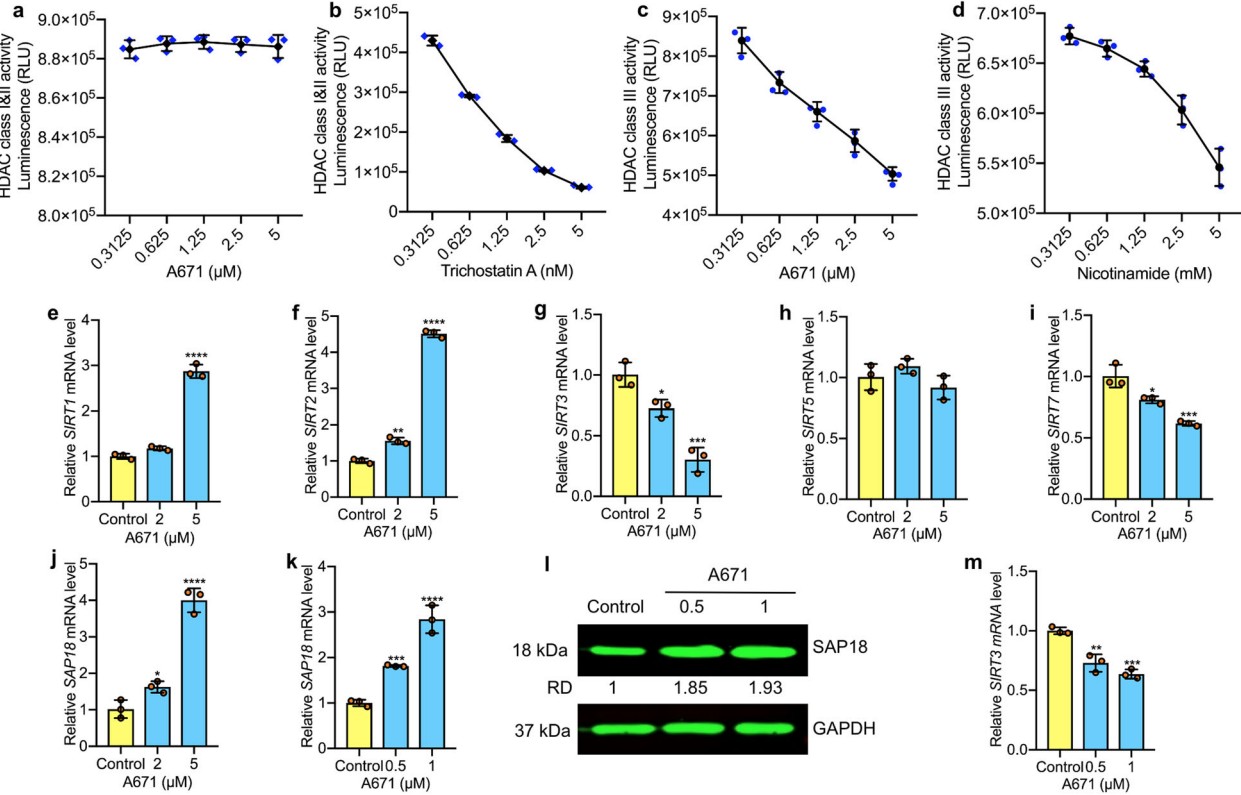

**Fig. 3 A671 is an HDAC type III inhibitor (HDACi-III) that attenuates SIRT3 by binding and activating SAP18. a** HEL cells incubated for 30 min with the indicated concentrations of A671 followed by HDAC type I and IIA/IIB activity assays (Promega). **b** Trichostatin A used as a positive control. **c** HEL cells were treated with A671, followed by HDAC type III activity analysis. **d** Nicotinamide was used as a positive control. **e–j** Q-RT-PCR analysis of EL4 cells for the indicated genes following 12 h treatment with A671, DMSO, or medium. **k, l** Induction of SAP18 mRNA (**k**) and protein (**l**) levels by A671 in a dose-dependent manner. **m**, A671 treatment diminished *SIRT3* transcription in a dose-dependent manner as determined by Q-RT-PCR. $P < 0.05$ (*), $P < 0.01$ (**), $P < 0.001$ (***), $P < 0.0001$ (****) by two-tailed student *t*-test.

induction of *SAP18* (3.4-folds, $P = 0.013$), a subunit of the tumor suppressor histone deacetylase complex, SIN3[17]. We therefore further analyzed the effect of A671 on SAP18 and SIRT3 via Q-RT-PCR. *SAP18* mRNA induced in a dose-dependent manner following exposure to A671 in EL4 cells (Fig. 3j). This negative correlation between *SAP18* induction and *SIRT3* downregulation (Fig. 3m) was also seen both at the mRNA and protein levels in A671-treated HEL cells. In contrast, MDA-MB-231 and 4T1 cells, which are insensitive to A671 (Fig. 1b), exhibited no change in expression of SIRT3 or SAP18 (Supplementary Fig. 2a, f), in response to this drug.

The aforementioned results suggest that the SIN3-SAP18 complex may underlie transcriptional suppression of *SIRT3* and *SIRT7* by A671. To examine this possibility, we silenced SAP18 expression via shRNA-SAP18 (Sh-SAP18) in HEL cells and tested the effect on SIRT3 expression (Fig. 4a, b). Remarkably, SAP18 silencing resulted in over a 3-fold induction in *SIRT3* expression (Fig. 4c). The effect was specific to *SIRT3*; *SIRT1*, *SIRT2*, and *SIRT7* were not affected by SPA18 depletion (Supplementary Fig. 3). Next asked whether over-expression of SAP18 would mimic A671 leading to suppression of *SIRT3*. HEL cells transduced with a SAP18 expressing lentivirus and over-expression of SAP18 confirmed by immune-staining (Supplementary Fig. 4a), and Q-RT-PCR analysis (Supplementary Fig. 4b). Importantly, the expression of *SIRT3* has significantly suppressed in *SAP18* over-expressing cells compared to control (Supplementary Fig. 4c).

The depletion of SAP18 in shSAP18-HEL cells had little effect on cell proliferation compared with control scramble shRNA

(Fig. 4d). Yet, shSAP18-HEL cells were significantly more resistant to A671 relative to scrambled control cells (Fig. 4e), pointing to SAP18 as a potential target for A671. Moreover, overexpression of SIRT3 significantly increased cell proliferation (Supplementary Fig. 5a, b) and conferred resistance to A671 treatment (Supplementary Fig. 5c), indicating that SIRT3 down-regulation is a major mechanism by which A671 exerts its inhibitory effect.

The induction of cell death in response to A761 and its impact of SAP18/SIN3 and SIRT3 raised the possibility that the latter genes may control cell survival. A search of the DepMap CRISPR database of gene knockout for SIRT3/SAP18/SIN3 across hundreds of cancer cell lines identified SAP18/SIN3 but not SIRT3 as a critical survival gene, (Supplementary Fig. 6). To determine the effect of SIRT3 depletion in HEL cells, we assessed several shRNAs (sh-SIRT3-1-3) and identified sh-SIRT3-2 as a potent silencer of SIRT3 (Supplementary Fig. 7a–c). Sh-SIRT3-2-mediated knockdown of SIRT3 resulted in a significant decrease in cell proliferation without an obvious impact on cell survival (Supplementary Fig. 7d).

The aforementioned results suggest that A671 activates SAP18 and thereby the SIN3 repressor to shut down SIRT3 expression. To directly test this possibility, we asked whether SIN3 is recruited to the SIRT3 promoter using publically available Chromatin Immunoprecipitation (ChIp) data from ENCODE on two cell lines: GM12878 (B-lymphocyte-lymphoblastoid) and H1-hESC (human embryonic stem cells; www.encodeproject.org). Remarkably, SIN3A recruited with the highest affinity to the *SIRT3* gene promoter (Supplementary Fig. 8). Affinity of SIN3A

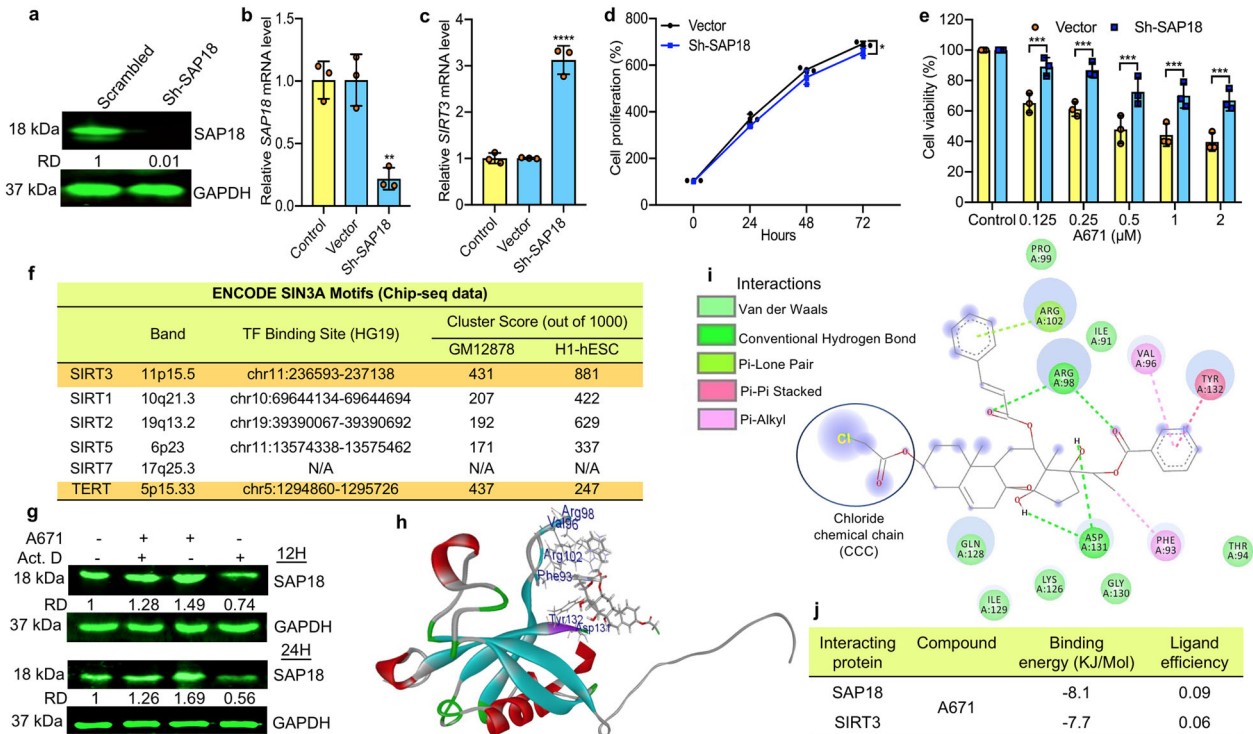

**Fig. 4 A671 binds SAP18 within the SIN3 suppressor complex to transcriptionally repress *SIRT3* expression. a, b** Efficient depletion of SAP18 in HEL cells using shRNA as determined by western blotting (**a**) and Q-RT-PCR (**b**). Untreated or scrambled transfected vector (Vector) cells used as control. **c** Depletion of SAP18 in HEL cells (Sh-SAP18) induces *SIRT3* transcription. **d** Marginal effect of Sh-SAP18 on cell proliferation. **e** Survival rate of Sh-SAP18 and controls cells treated for 24 h with indicated concentrations of A671. **f** SIN3A transcription factor (TF) binding sites within the Sirtuin gene family promoters. DNA sequencing data derived from SIN3A chromatin immunoprecipitation (ChIP) for two cell types (GM12878, B-lymphocyte lymphoblastoid, and H1-hESC, human embryonic stem cells) obtained through the ENCODE database (for details see supplementary Fig. 8). Of the Sirtuin genes, SIN3A observed to bind within the promoters of all but *SIRT7*. The greatest binding observed for *SIRT3* was ~108% higher than *SIRT1* in GM12878 cells and ~40% higher than *SIRT2* in the H1-hESCs. **g** Expression of SAP18 in HEL cells treated with actinomycin D (10 μM) alone or in combination with A671 (0.5 μM) for 12 (top) or 24 h (bottom). GAPDH used as a loading control. **h** A three-dimensional view of the predicted interaction of A671 with SAP18. **i** Position of chemical interactions with the indicated amino acids within SAP18. **j** Predicted affinity properties of A671 to SAP18 and SIRT3. $P < 0.05$ (*), $P < 0.01$ (**), $P < 0.001$ (***), $P < 0.0001$ (****) by two-tailed student $t$-test.

to the *SIRT3* promoter was as high or even higher than that observed for the *TERT* promoter, a known SIN3A regulated gene (Fig. 4f)[33]. SIN3A showed a much lower affinity to *SIRT1*, *SIRT2*, *SIRT5*, and not at all to *SIRT7*, suggesting that the *SIRT3* promoter may be directly and uniquely regulated by the SIN3/SAP18 complex (Fig. 4f).

**Auto-regulation of SAP18 and its stabilization by A671.** To further analyze the effect of A671 on SAP18, we treated HEL cells with the transcription inhibitor actinomycin D in the presence or absence of A671. Actinomycin D or A671 alone suppressed or induced SAP18 expression, respectively, whereas combined A671 plus actinomycin D treatment resulted in sustained SAP18 expression comparable to control DMSO treated cells (Fig. 4g). Thus, A671 counteracts the effect of actinomycin D, suggesting transcription repression by A671. Increased SAP18 protein expression by A671 was slightly enhanced by the proteasome inhibitor MG132 (Supplementary Fig. 9), suggesting that A671 may directly bind to and stabilize SAP18 in leukemic cells. Since *SAP18* mRNA and protein are both elevated in A671 treated cells (Fig. 3j–l), the A671–SAP18 interaction may elicit transcriptional activation of the latter. In accordance, the protein synthesis inhibitor cycloheximide (CHX) was downregulated *SAP18* mRNA and strongly induced *SIRT3* transcription (Supplementary Fig. 10a–c).

**Docking analysis.** This predicted direct binding of A671 to SAP18, leading to the induction of its repression activity. We next asked whether A671 induced SAP18 protein expression through direct interaction with this SIN3 subunit. Docking analysis identified a strong affinity between A671 and SAP18 (Fig. 4h, i) compared to SIRT3 (Supplementary Fig. 11a–c). Indeed, A671 predicted to have a binding energy of −8.1 KJ/Mol and ligand efficiency of 0.09 for SAP18 compared to −7.7 KJ/Mol and 0.06 for SIRT3 (Fig. 4j).

SAP18 contains a ubiquitin-like fold domain with several large loop insertions[34]. Our docking analysis predicted that A671 interacts with one of these loops between amino-acids 98–132 (Fig. 4h, i). A671 formed stronger hydrophobic, Pi Lon Pair, and van-der-Waal bonds with ten amino-acids within this loop (Fig. 4i, green). Also, A671 predicted to form weaker Pi–Pi stalked and Pi-Pi alkyl bonds with three amino-acids, A132, A96, and A93 (Fig. 4i, purple). When bound to A671, the Chloride Chemical Chain (CCC) (Fig. 4h) not predicted to contact SAP18, but the activity of the SAP18/SIN3 complex may still be induced in sensitive cells, leading to cell death. This inhibitory activity is in contrast to two similar compounds (A670 and A672), having different chemical bound substitution (Supplementary note).

To further confirm the binding of A671 to SAP18, pool-down experiments performed using epoxy-activated Sepharose 6B (ES6B) beads[35,36]. Extracts isolated from HEL cells incubated with conjugated A671 + ES6B or ES6B beads alone, and after

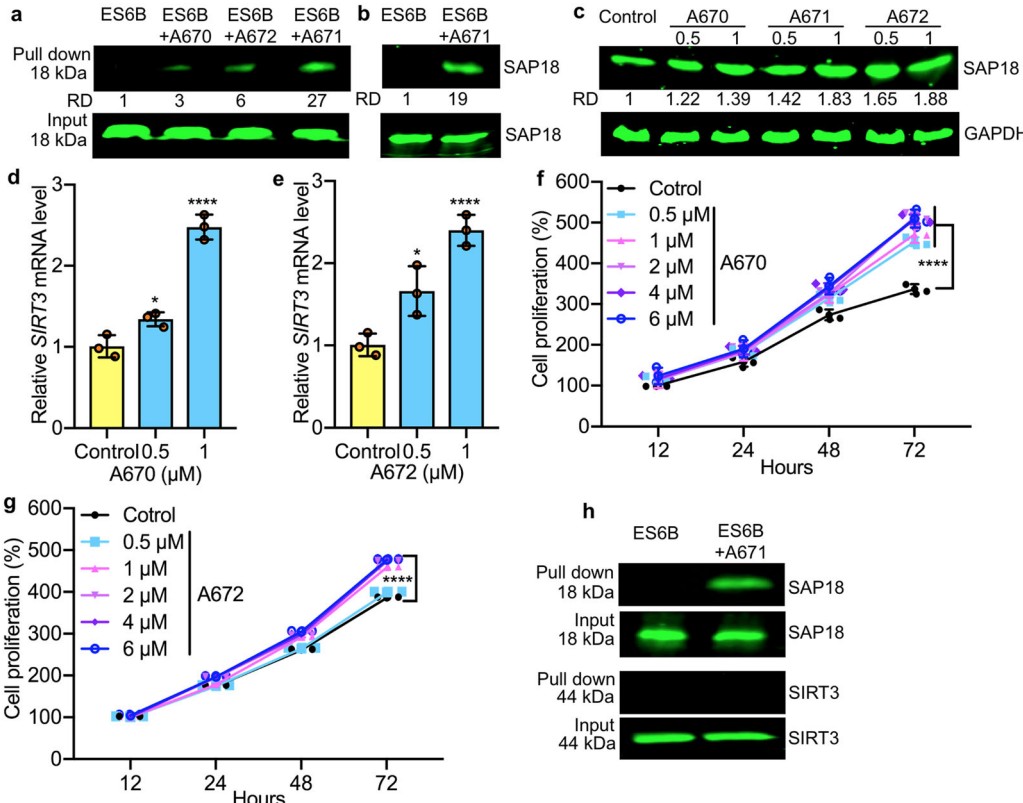

**Fig. 5 Binding of A671 and related compounds to SAP18. a** Western blotting for SAP18 following pull-down of Sepharose 6B beads loaded with indicated compounds. Sepharose 6B beads without compound used as control. **b** Binding of A671 to recombinant SAP18 in pull-down assays. **c** Western blot for SAP18 expression 12 h after treatment with indicated concentrations of A670-A672 compounds. **d**, **e** Expression of SIRT3 in HEL cells treated for 24 h with indicated concentrations of A670 (**d**) or A672 (**e**). **f**, **g** Growth curves of HEL cells treated for indicated periods and concentrations of A670 (**f**) or A672 (**g**). **h** Pull-down assay for binding of A611 to SAP18 and SIRT3 in HEL cells. The input used as a positive control. $P < 0.05$ (*), $P < 0.0001$ (****) by two-tailed student $t$-test.

washing, the eluted proteins subjected to western blotting. A band corresponding to SAP18 was observed in ES 6B-A671-treated cells but was absent in control ES6B alone (Fig. 5a). This affinity binding was then confirmed using recombinant SAP18 protein in which a specific band precipitated by ES6B + A671 (Fig. 5b).

Compounds A670 and A672, which lack growth inhibitory activity, also precipitated SAP18, albeit with much lower affinity compared to A671 (Fig. 5a). Both A670 and A672 also increased the stability of SAP18 in HEL cells (Fig. 5c), but the contrast to A671, they upregulated SIRT3 expression in a dose-dependent manner resulting in higher proliferation in culture (Fig. 5d–g compared with A671, Fig. 1c). Thus, specific interactions of these structurally related molecules have distinct impacts on SAP18 and the SIN3 complex.

While A671 was able to pull down SAP18, it did not precipitate SIRT3 in a side-by-side experiment (Fig. 5h). As SAP18 is a unit of the SIN3 complex[16], we also detected co-precipitation of SIN3A protein in A671 (Supplementary Fig. 12), but not in A670 and A672 pull-down experiments. Together, these results suggest that A671 stabilizes SAP18 through direct binding, which, through the CCC domain, increases its inhibitory effect on SIN3 complex, leading to transcriptional suppression of *SIRT3* and cell death (Fig. 6a).

### Correlation between SIRT3 and SAP18 expression in leukemic patients.
As noted, the effect of A671 on SAP18 and SIRT3 was cancer-specific (Fig. 1b). For example, A671 had no inhibitory effect on cell growth (Fig. 1b) or SAP18 mRNA or protein expression (Supplementary Fig. 2c–f) in the breast cancer cell

lines MDA-MB-231 and 4T1. Thus, the ability of A671 to stabilize SAP18 and to block SIRT3 expression is a potential biomarker of tumor responsiveness to the drug.

The negative correlation between SAP18 and SIRT3 may be characteristic of other malignancies and may indicate sensitivity to A671. We examined this possibility using an mRNA expression dataset of 301 samples from 192 patients with B-cell acute lymphoblastic leukemia/lymphoma (BCALL), extracted from a total of 1978 patients with pediatric acute lymphoid leukemia. We detected a significant negative correlation between *SIRT3* and *SAP18* in BCALL patients, with a Rho ($r$-value) of $-0.4022$ and a 95% confidence interval (CI) of $-0.4954$ to $-0.2998$ (Fig. 6b). There was also a strong association between high *SAP18* and *SIRT3* levels with better or worse prognosis, respectively (Fig. 6c, d). Interestingly, the BCALL cell line OCI_LY10 and another B-cell lymphoblastic cell line Raji also exhibited high sensitivity to A671 (Fig. 1b). These results point BCALL as a strong candidate for A671-based therapy.

## Discussion
Current treatments for T-cell lymphomas include combinations of chemotherapy, radiotherapy, and surgery[6–8]. However, 20–30% of patients with non-Hodgkin lymphoma (NHL) recur after initial therapy[6]. Targeted therapy urgently needed for refractory patients that failed chemo and/or radiotherapy. Three HDAC inhibitors have been FDA-approved for treating T-cell lymphomas[10–13]. Here, we identified a unique HDACi that possessed a potent anti-T cell leukemia activity by direct binding to and stabilization of SAP18, leading to downregulation of *SIRT3*

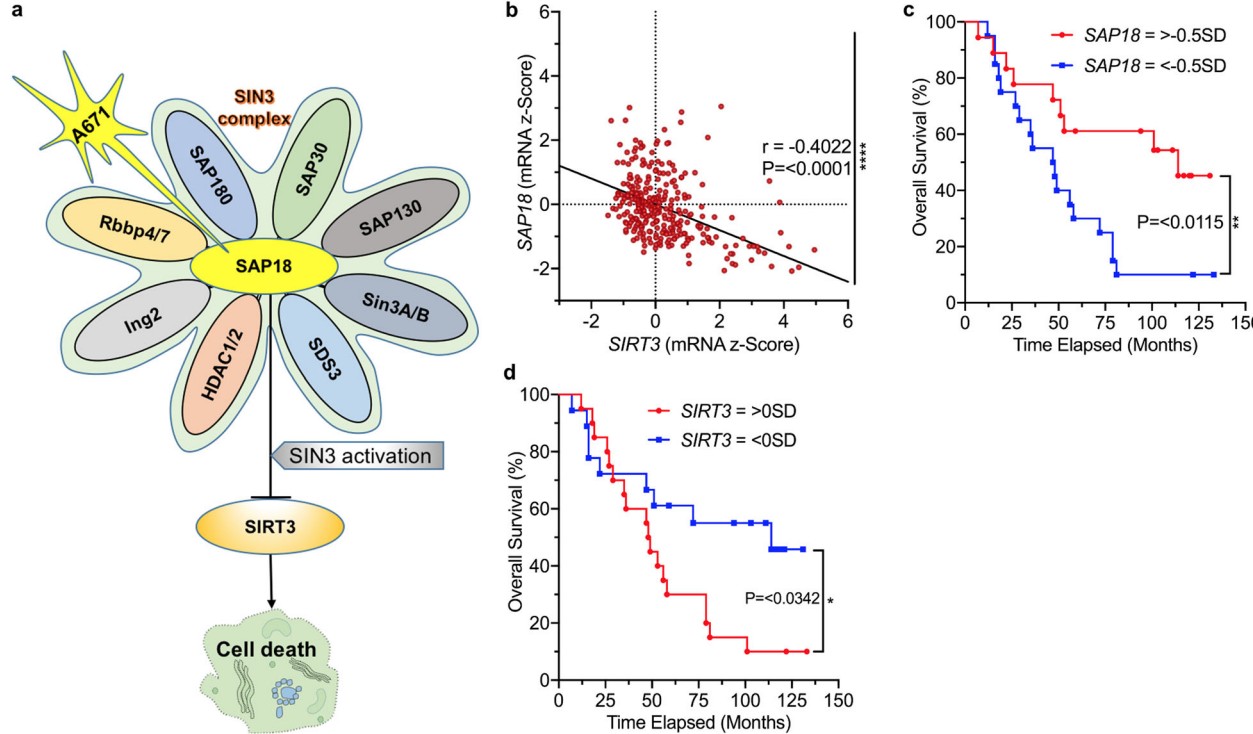

**Fig. 6 Correlation between SAP18 and SIRT3 transcription in BCALL patients predicts drug response. a** A model for the effect of A671 on SAP18/SIN3 and SIRT3, and its impact on leukemic cell survival. A671 binds and stabilizes SAP18, leading to activation of the SIN3 complex and transcriptional repression of *SIRT3* expression, which culminates in apoptotic cell death. **b** A significant negative correlation ($P = 0.0001$ by two tailed student $t$-test) of SIRT3 and SAP18 in patients with BCALL, and correlation ($r$) of 0.4. (Spearman's rank analysis). **c, d** Overall survival rates for high SIRT3 and SAP18 expression in BCALL patients determined by Log-rank (Mantel-Cox) test. < −0.5 SD (less than −0.5 standard deviations below the mean) had worse overall survival when compared to patients whose SAP18 with higher levels (> −0.5 SD). > −0.5 SD $n = 18$, < −0.5 SD $n = 20$; Median survival > −0.5 SD 114 months and < −0.5 SD 47.5 months; Log-rank hazard ratio (> −0.5 SD/< −0.5 SD) = 0.3800; 95% CI of ratio = 0.1774–0.8137; $P = 0.0115$ (**). The above mean (>0 SD) expression of SIRT3 (compared to SIRT3 < 0 SD) is associated with a significantly lower overall survival rate for patients with BCALL. >0 SD $n = 20$, <0 SD $n = 18$; Median survival = > 0 SD 48.5 months and <0 SD 114 months; Log-rank hazard ratio (>0 SD/<0 SD) = 2.277; 95% CI of ratio = 1.069–4.849; $P = 0.0342$ (*) by two tailed student $t$-test.

transcription through activation of the SAP18/SIN3 suppressor complex, resulting in apoptotic cell death. The compound A671, with a new mode of anti-leukemic action, should then be assessed in the clinic for treatment of drug-refractory lymphomas/leukemias.

A671 induced-apoptosis is associated with downregulation of several pro-survival genes, including BCL2, phospho-ERK, and AKT, whereas other factors such as cMYC and, FLI1 are unaffected. This observation is consistent with a previous report using a different HDACi[37]. SAP18 reported affecting macrophage differentiation through inhibition of STAT3 phosphorylation and ERK[38]. While SIRT3 knockdown resulted in significant inhibition of proliferation, other SAP18 targets may cooperate with this sirtuin to exert cell death by A671. Future identification of additional SAP18 target(s) and demonstration of their direct binding to promoters of these genes would further our understanding of the mechanism of A671-induced cell death.

Our study demonstrates that A671 induces apoptosis in selected cancer cells, predominantly T-cell lymphoma, erythroleukemia, and selected B-cell lymphomas. While SAP18 expression induced in A671 sensitive tumor cells, its transcript and protein levels remained unchanged in drug-resistant cells. Since the binding of A671 to SAP18 is critical for its effects, it is possible that this compound–target interaction not accessible in drug-resistant cancer cell lines. Indeed, a recent study by Qu et al.[9] demonstrated that the clinical response of cutaneous T-cell

lymphoma (CTCL) to HDACi strongly associated with a concurrent gain in chromatin accessibility compared to T cells from these patients and normal individuals.

A molecular docking combined with pull-down immunoprecipitation experiments revealed binding of A671 as well as two related compounds (A670 and A672) to SAP18, leading to higher levels of expression through protein stabilization. However, only A671, which contains a CCC domain, blocked SIRT3 expression through higher binding affinity to SAP18 and activation of the SIN3 complex. Derivatives (A670 and A672), which are devoid of CCC, also bind SAP18 and induce protein stability, but they induced SIRT3 expression and cell survival, likely through inhibition of SIN3 activity. Thus, the presence of CCC appears to dictate whether these drugs act as antagonists or agonists. Although in our system SIRT3 functions as a pro-survival protein, anti-survival activity of this sirtuin were reported in other cancer cells[21,22,39]. This dual oncogenic and tumor suppressor activity may stem from distinct SIRT downstream signaling in different cell types, an intriguing observation that may require further investigation.

Interestingly, while A671 binds to SAP18 and increases protein expression through stabilization, this compound also increases SAP18 transcription, possibly through a non-SIN3 complex, which remained to be defined in future studies. SAP18 interacts with TRIB1 to regulate the expression of the MTTP gene[40], indicating that this SIN3 component can exerts both positive and negative effect in different contexts.

Administration of A671 in animal models inhibited T-cell lymphoma and erythroleukemia, which are resistant to standard therapeutics[41,42]. The ratio between *SAP18* and *SIRT3* predicted the sensitivity of BCALL to the A671 in TCGA and cell line analysis. Also, *SIRT3* highly expressed in Diffuse large B cell lymphomas (DLBCLs), which are dependent on mitochondrial lysine deacetylase SIRT3 for proliferation, survival, self-renewal, and tumor growth[20]. Thus, A671 is a better candidate drug for the treatment of both DLBCLs and BCALL patients. The impact of these compounds on SAP18 stability and the SAP18/SIRT3 ratio may serve as indicators for survival and drug response in other cancer types.

In summary, we identified the $C_{21}$-steroidal derivatives with agonist and antagonist effects on the SAP18/SIN3 complex by targeting *SIRT3* in leukemic cells. The SAP18 agonist compound, A671, with potent inhibitory effect against T-cell lymphoma and erythroleukemia through *SIRT3* downregulation needs further investigation for the treatment of these, and other malignancies in which the SAP18/SIRT3 regulatory axis is intact.

## Methods

**Cell lines**. Verified mycoplasma negative tested human and murine cell lines originated from T-cell lymphoma (EL4, CEM-C7H2, MOLT4, BW5147), erythroleukemia (CB3, CB7, DP17, K562, HEL), human embryonic kidney (HEK293T), B-cell lymphoma (Raji, OCI_LY10, TK6, and BJAB), Multiple myeloma (RPMI-8226), Melanoma (WM9, WM239), breast cancer (MDA-MB-231, 4T1, MCF7, FE1.2, and FE1.3) and prostate cancer (PC3, BPH) were maintained in Dulbecco's Modified Eagle Medium supplemented with 5% fetal bovine serum (HyClone, GE Healthcare, Australia).

**Tumor induction and in vivo compound treatment**. To induce erythroleukemias, newborn BALB/c mice injected intraperitoneally (IV) with F-MuLV[24,43]. At five weeks of post-viral infection, leukemic mice have injected IV every other day for a total of seven injections with A671 compound (3 mg/kg body weight) and as control with vehicle control DMSO. Leukemic and drug-treated mice then monitored for the development of erythroleukemia. Mice with signs of late-stage disease were sacrificed by dislocation by experienced staff and used to determine survival (%), hematocrit level, and tumor weight[24]. For T-cell lymphoma induction, EL4 cells ($5 \times 10^6$) were injected IV into C57BL/6 mice (2 months old) and were treated every other day with A671 (3 mg/KG body weight; seven times) a week after the initial injection. The mice subsequently monitored for leukemia development. Survival rates of animals were computed and plotted according to the nonparametric Kaplan–Meier analysis.

**Drug screening studies, IC$_{50}$, cell cycle, and apoptosis analysis**. Cancer cell lines ($1 \times 10^4$) were plated in triplicates into 96-wells plates and incubated with various concentrations of compounds. After drug treatment, cells were subjected to Tetrazolium dye (MTT) assay[43]. The IC$_{50}$ values (concentration of compound required to reduce 50% of cell viability) was calculated accordingly[43].

The cell cycle and apoptosis conducted with published methods[43]. In brief, cells treated with compounds for 24 h. For apoptosis detection, cells ($1 \times 10^6$ cells) were washed twice with cold PBS and resuspended in 1 ml of $1\times$ binding buffer. Then transferred 100 μl of the solution ($1 \times 10^5$ cells) to culture tubes (2 ml) and stained by FITC Annexin V (5 μl)/PI (5 μl) apoptosis detection kit (BD Biosciences, Franklin Lakes, NJ). The cells gently mixed by vortex, incubated for 15 min at RT (25 °C) in the dark, then 1X Binding Buffer (400 μl) was added into each tube and analyzed by flow cytometry within 1 h. For cell cycle analysis, cells were fixed by cold 75% ethanol overnight at −20 °C, washed with cold PBS, stained in PI for 40 min at 25 °C, then analyzed by NovoCyte flow cytometer (ACEA NovoCyte 2040R, USA).

**Western blotting and inhibitory compounds**. Western blots performed according to published methods[43,44]. Polyclonal rabbit antibodies for FLI1, BCL2, SIRT3, SIN3A, cMYC obtained from Abcam, Cambridge, UK; ERK, phospho-ERK, AKT, phospho-AKT, GAPDH, SAP18 antibodies obtained from Cell Signaling Technology (CST), Danvers, MA01923; goat-anti-mouse and goat anti-rabbit HRP-conjugated antibodies obtained from Promega, Madison, Wisconsin, USA. Antibody dilution conducted according to the manufacturer's instructions. The Oddessy system (Li-Cor Biosciences, Lincoln, USA) used for protein detection. Proteasome (MG132) and transcription (Actinomycin D) and translation (Cycloheximide) inhibitors were obtained from Selleckchem.com and used in some experiments.

**Total RNA-cDNA isolation and quantitative real-time PCR**. Total RNA was extracted from cells using TRIzol (Invitrogen Life Technologies, Carlsbad, USA)[43]. PrimeScript RT Reagent kit obtained from Takara (Takara, Dalian, China) was used for cDNA isolation and reverse transcription reactions. Quantitative real-time PCR (Q-RT-PCR) performed by FastStart Universal SYBR Green Master kit (Roche, Mannheim, Germany) and the Step One Plus Real-time PCR system (Applied Biosystems, Singapore, Singapore). The expression of each gene calculated relative to GAPDH using the $2{-}\Delta\Delta Ct$ method. Primer sequences for PCRs listed in Supplementary Table 1. All PCR experiments performed in triplicate for at least three biological replicates.

**shRNA cloning, expression vectors, and lentiviral infection**. The SAP18 and SIRT3 shRNA construct was generated through synthesis by Vigene Bioscience, Shandong, China, and cloned into the BamHI/M1uI sites of the PLent-U6-GFP-Puro-SAP18 and PLent-GFP-Puro-SIRT3 plasmids (Vigene Bioscience, Rockville, MD, USA). The sh-SAP18 and sh-SIRT3 sequences provided in Supplementary Table 2. For lentivirus production, the sh-SAP18 or sh-SIRT3 expression plasmid, packaging plasmids psPAX2 and pMD2.G (a gift from Didier Trono, Addgene plasmid #12260 and #12259) were co-transfected into HEK293T cells[43]. The supernatants were harvested 48 h post-transfection and centrifuged at 1000 g for 10 min, filtered through 0.45 μm filters and of the virus-containing supernatants were added to HEL cells for transduction. The culture medium was changed 24 h post-transduction and cells grown in the presence of puromycin (Solarbio, Beijing, China) until puromycin-resistant cells obtained.

Cells were transfected using Lipofectamine 3000 transfection reagent (Thermo Fisher Scientific-US) with either pcDNA3.1 containing SIRT3 cDNA (pcDNA3.1hSIRT3_H248Y_HA; Addgene, USA) or an empty vector plated onto a 6-wells plate. The transfected cells were selected with 1.8 mg/ml neomycin (G418; Sigma Aldrich, USA) for two weeks and expanded with 0.5 mg/ml neomycin. SIRT3 overexpression assessed by Western blot[45]. Similarly, SAP18 (pCMV3-SAP18-GFPSpark®; Sino Biological, China) overexpression studies conducted using Lipofectamine 3000 and hygromycin (250 μg/ml) used for cell selection. Details of the plasmids used for this study provided in Supplementary Table 3.

**HDAC I/II and SIRT3 cell-based assays**. HDAC I&II and SIRT3 inhibitor potency determination were analyzed using intact HEL cells[46,47]. Briefly, for HDAC I&II experiment, an initial dilution of A671 compound and the known HDAC inhibitor Trichostatin A prepared with serial two-fold or three-fold dilutions in a white-walled 96-well plate, according to Promega kit instructions (Promega, CAT.G6430). Fifty microliter of the HDAC enzyme dispensed into each well, and the enzyme/inhibitor mixes incubated at room temperature for 30 min. Hundred microliter of the developer reagent and an equal volume (100 μl) of HDAC-Glo™ I&II reagent added into each well. The plates mixed at room temperature before the luminescence at a signal steady-state was determined. Nicotinamide used as a positive control for the determination of the SIRT3 activity.

**Pull down immunoprecipitation**. The compounds (A670, A671, and A672) around 1 mg dissolved in 1 ml of coupling buffer (0.1 M of NaHCO$_3$ containing 0.5 M of NaCl; pH 11). The epoxy-activated Sepharose-6B (Sigma Aldrich, USA) beads washed in distilled water followed by the coupling buffer and coupled to the compounds during an overnight incubation at 4 °C. After which, the beads washed, the unoccupied binding sites blocked with Tris-HCl buffer (0.1 M; pH 8) for two hours at room temperature. The conjugated compounds washed with three alternating cycles of pH wash buffers I (0.1 M of acetate and 0.5 M of NaCl, pH 4) and II (0.1 M of Tris-HCl and 0.5 M of NaCl, pH 8). The control unconjugated epoxy-activated Sepharose 6B beads were prepared similarly in the absence of compounds. Cell lysates were mixed with the compound conjugated Sepharose 6B beads at 4 °C overnight[35] and subsequently washed once with TBST. The bound proteins were eluted with SDS loading buffer and resolved by SDS-PAGE tailed by immunoblotting with antibodies against SAP18. Recombinant SAP18 was obtained from Abcam and used in affinity binding assays to test for interaction of A671 with SAP18.

**Computation docking analysis**. The three-dimensional structure of A671 was analyzed and drawn in ChemSketch. The protein crystallographic structures of receptors SAP18 (PDB ID: 2hde) and SIRT3 (PDB ID: 3GLS) retrieved from www.rcsb.org. Auto Dock tools 1.5.6.[48,49] used to compute the molecular docking simulations following the standard protocol, as described in the software documentation. Furthermore, the interacting sites were analyzed using 2D lig plot analysis. The three-dimensional structures of A671 predicted and drawn in ChemSketch.

**Animal care**. Animal care and procedures followed by the criteria for the use of laboratory animals. The animal protocol was reviewed and approved by the Guizhou Medical University Animal Care Committee under the guidelines of the China Council of Animal Care (Approval ID #1900373).

**Statistics and reproducibility**. Statistical analysis was performed using the two-tailed student t-test with significance denoted by *$P < 0.05$, **$P < 0.01$, ***$P < 0.001$ and ****$P < 0.0001$. The analysis of variance performed using Origin 3.5 software (Microcal Software, Northampton, MA, USA).

The publicly available pediatric acute lymphoid leukemia dataset from phase 2 of the NIH/NCI (National Institutes of Health/National Cancer Institute, USA) TARGET initiative was downloaded from cBioportal[50,51] to produce a dataset containing the mRNA expression data (z-scores, where the gene expression for each patient described as a standard deviation from the mean expression of the cohort for each gene) for SAP18 and SIRT3. The expression data subsequently merged with the clinical data available on cBioportal, which enabled the cross-referencing of the expression survival data and was available for 110 of the 301 samples. Additionally, patients with more than one sample recorded the database were excluded for the survival analysis (leaving 38 of the 110 samples). The statistical analysis of the TARGET data conducted using Graphpad Prism software (version 8.1.1), as detailed in the applicable figure legends and for ChIp analysis, ENCODE-database (WWW.encodeproject.org)[52,53] was used to identify factors bind to the sirtuin promoter genes. The database WWW.depmap.org used to identify dependency for genes SIRT3, SAP18, and SIN3.

**Reporting summary**. Further information on research design is available in the Nature Research Reporting Summary linked to this article.

## Data availability
Raw data for graphs can be found in Supplementary Data 1. Full blots and gating strategy for flow cytometry data are provided in Supplementary Information. All other data are available within the manuscript files or from the corresponding author upon reasonable request.

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

## Acknowledgements

This work was supported by research grants from The Natural National Science Foundation of China (21867009, 31760529, and U1812403), the Science and Technology Department of Guizhou Province innovation and project grants (6012, 4001, QKHPTRC [2019]5627, QKH-ZC[2019]2754, QKH-PTRC[2017]5101, QKH-PTRC[2017]5737, QKH-PTRC[2018]5779-61) and the 100 Leading Talents of Guizhou Province to YBD and HXJ.

## Author contributions

B.G., W.L., C.W., and K.M.S. contributed to the conception, design of the study as well as data acquisition and interpretation. C.J.W. contributed to human BCALL analysis. B.G. drafted the manuscript. K.M.V. performed docking analysis. L.H. and X.H. generated compounds. Y.B., K.M.S., X.H., C.J.W., and E.Z. reviewed the manuscript critically. Y.B. supervised and designed the study. All authors contributed to interpretation of findings, reviewed, edited, and approved the final manuscript.

## Competing interests

The authors declare no competing interests.
