## [Peer Review File · Communications Biology]

Reviewers' comments:

Reviewer #1 (Remarks to the Author):

In the presented study, Gajendran and colleagues report the identification of a novel anti-leukemic compound, A671, and profile its effects in vitro as well as using in vivo animal models. The authors demonstrate an induction of SAP18 expression by A671 and correlate this to a decrease in SIRT3 expression. Moreover, they propose a direct interaction between A671 and SAP18, suggesting this protein is the predominant biochemical target of A671. While the work is of interest, there are minor issues with publishing the work in its present form:

1) Based on the provided data, I do not believe there is sufficient evidence to claim that SAP18 is a direct target of A671. While the authors have shown a strong correlation between the activity of A671 and SAP18 levels, there is no empirical data to suggest a direct molecular interaction outside of docking simulations. Although the pulldown experiments in figure 5 are compelling, using crude mammalian cell lysate as a source of SAP18 does not provide sufficient purity to rule out the possibility of SAP18 indirectly binding to A671 through a larger protein complex. This concern may be relieved by simply repeating the pulldown assay using recombinantly produced, purified SAP18 or using this recombinant SAP18 to do another direct binding assay such as isothermal titration calorimetry, biolayer interferometry, etc. As such, the language in the manuscript should be altered to reflect this lower level of certainty or the experiment should be performed.

2) Based on the provided data, I do not believe there is sufficient evidence to claim that SAP18 directly regulates levels of SIRT3 in response to A671. The authors show a decrease in SIRT3 expression in response to A671, as well as a concomitant increase in SAP18 expression. Additionally, the authors show ChIP-seq. data revealing an enrichment of SIN3 at the SIRT3 promoter. While these events are certainly correlated, it is improper to claim that SIRT3 is directly regulated by SAP18 in response to A671 without a ChIP-seq. experiment treating cells with A671 and then showing an increase of SAP18 at the SIRT3 promoter relative to untreated cells. As such, the language in the manuscript should be altered to reflect this lower level of certainty or the experiment should be performed.

3) that the authors need to tone down the language used when discussing the molecular mechanisms of A671 action. Specifically, while it is appropriate to say that A671 is a novel anticancer compound, I do not believe it is appropriate to draw conclusions about specific molecular targets without further experimental evidence other than to say that A671 is an inhibitor of class III HDACs. Moreover, while there are certainly changes in the expression of multiple HDACs in response to A671, it is not appropriate without further experimental evidence to suggest a specific protein (SAP18) is directly responsible for this. If the authors wish to keep the text of the manuscript as it is now, they should provide more direct evidence of their proposed biochemical mechanisms.

Reviewer #2 (Remarks to the Author):

Summary

The authors have shown a novel anticancer therapeutic, A671, which appears to stabilize the SIN3 complex by binding SAP18. The authors show one transcriptional consequence of this mechanism is the suppression of SIRT3, and attribute the anticancer activity of A671 to this reduction in SIRT3 abundance.

Overall comments

The manuscript is well written, and the impact on the overall cancer burden in mice is especially compelling that this molecular strategy could be a useful addition to our armament of anticancer drugs. The mechanism is interesting, but is so far largely based on biased (literature informed) experiments, and lacks any genome-wide data that would give a more compelling and complete

view of the molecule's mechanism. Many experiments are very well done – I especially love the rescue experiment in Fig 4e showing shRNA of SAP18 attenuates the activity of A671, a critical and clear experiment that argues a strong link between their proposed mechanism and the anticancer effect of the drug. Overall, I support publication once a few key concerns are addressed.

Major points

1. SIN3, a complex containing HDAC1/2, is referred to here only as a repressive complex. However, important new insights in this field are shifting this view. For instance, HDAC1/2 (and HDAC3) are all bound in the epigenome at the strongest enhancers and at the promoters of the most actively transcribed genes (2009, Cell, Wang, Z. et al. Genome-wide Mapping of HATs and HDACs Reveals Distinct Functions in Active and Inactive Genes and 2019, Nature Genetics, Gryder, B. E. et al. Histone hyperacetylation disrupts core gene regulatory architecture in rhabdomyosarcoma). The authors need to provide some evidence rule out (or, rule in) the possibility that their drug A671 isn't disrupting a potential need of SIN3 for maintenance of the most active and most lineage specific genes. RNA-seq analysis +/- A671, followed by GSEA analysis showing gene sets that go up as well as those that go down would be illuminating here. Both general gene sets, as well as those specific for EL4 cells (<https://www.ncbi.nlm.nih.gov/geo/query/acc.cgi?acc=GSE66343>), could be analyzed. Direct comparison for Entinostat and Nicotinamide RNA-seq in EL4 cells would be even more exciting and informative for mechanism. It is possible that SAP18/SIN3 has either activating or repressive roles depending on genomic context, and this consideration needs to be included. If the authors believe that the SIN3/SAP18 complex is more repressive than the other HDAC1/2 containing complexes (such as NuRD) that have been shown to be required for active gene expression (read and take into consideration the references above), then that too would be a helpful conclusion that RNA-seq data could offer.
2. Furthermore, this could hint at a more complete explanation of the way the drug downregulates Sirt3 while upregulating Sirt1/2 (which is itself very interesting). The relative expression is nice, but what is the absolute expression of Sirt1/2/3? If Sirt1/2 start at a low expression, and Sirt3 starts at a much higher level, then that can be thought of differently than if all three genes start at the same levels yet respond so differently. Again, getting transcripts per million from RNA-seq of the Sirt genes +/- A671 could be helpful by giving a sense of the absolute, not just relative, mRNA levels.
3. A central claim is that SIRT3 downregulation is responsible for the anticancer impact of A671. While there are experiments knocking down SAP18, I didn't see experiments where they knocked out SIRT3 and showed reduction in cell growth. I apologize if I missed this, but it would be essential to making the central claims of the paper.
4. The authors claim A671 is a HDAC-class III inhibitor. But this isn't true: it is a molecular glue for SAT8/SIN3. It is a HDAC-class III transcriptional suppressor, not an inhibitor.
5. A central claim is the "leading to recruitment of the SAP18-SIN3 complex to the SIRT3 promoter and its transcriptional silencing". Yet, this notable claim lacks any supporting data as far as I can see. To make this novel claim they need ChIP-seq or at least ChIP-qPCR for SAP18/SIN3 at the SIRT3 promoter (plus primers for a negative control region and primers for the SIRT1/2 promoters), +/- A671.

Novelty/notability claims

6. The papers novelty is high and the authors have followed a mostly well informed and courageous line of experimentation. I cannot find prior work that detracts from the overall novelty of this study.
7. The pulldown experiments are excellent additions that make the novel claims of direct binding to SAP18/SIN3 well supported.

Minor points

8. The interpretation of Fig 4f is oversold quite a bit. More careful analysis of actual ChIP-seq data would be needed to drawn such a strongly worded conclusion of gene specificity. Not to mention – promoter binding accounts for only a small fraction of functional binding events in the genome!

Differences in SIN3/SAP18 binding in enhancers could be even more important for gene-selective effects of the complex.

9. The authors make statements about the importance of Sirt3 to support tumor growth, and claim SAP18 at high levels improves outcomes by keeping SIRT3 low in human cancers. Can the authors make use of the DepMap CRISPR data for SIRT3/SAP18/SIN3 across the hundreds of cancer cell lines in that publicly accessible.

Figure comments

10. Figure 1a shows 3 molecules, but Fig 1b shows data from only 1 molecule. Help the reader by indicating in Fig 1b which molecule from Fig 1a is being tested (in the figure itself, not just in the legend). Same for Fig 1c. Fig 1d-1f all have A671 labeled so no need to change those.

11. Fig 1c please indicate cell line used in the figure itself, not just the legend. Same for Fig 1d-f.

12. Fig 2b-c have the same axis labels and the same conditions: please also indicate the tumor model so the difference between these two is apparent.

13. Figure 3a etc.: please indicate on the figure itself that this is an HDAC activity assay. Just add "HDAC type X activity" to the Y-axis label, above "Luminescence". Or, group a and b, and c and d, and have a hovering title for HDAC type activity over each pair.

14. In Figure 3 and others, many panels say "level". Can you additionally indicate in the figure axis labels either "mRNA level" or "protein level", whichever it is?

15. The model image needs a lot of work. Power point was used, but Adobe Illustrator would give the authors a much cleaning and eye-catching image. It's worth the investment, in my opinion.

6/8/2020

REF: COMMSBIO-20-1030A, A novel C₂₁-steroidal derivative suppresses T-cell lymphoma by inhibiting SIRT3 via SAP18-SIN3

Reviewers' comments:

Reviewer #1 (Remarks to the Author):

In the presented study, Gajendran and colleagues report the identification of a novel anti-leukemic compound, A671, and profile its effects in vitro as well as using in vivo animal models. The authors demonstrate an induction of SAP18 expression by A671 and correlate this to a decrease in SIRT3 expression. Moreover, they propose a direct interaction between A671 and SAP18, suggesting this protein is the predominant biochemical target of A671. While the work is of interest, there are minor issues with publishing the work in its present form:

1) Based on the provided data, I do not believe there is sufficient evidence to claim that SAP18 is a direct target of A671. While the authors have shown a strong correlation between the activity of A671 and SAP18 levels, there is no empirical data to suggest a direct molecular interaction outside of docking simulations. Although the pulldown experiments in figure 5 are compelling, using crude mammalian cell lysate as a source of SAP18 does not provide sufficient purity to rule out the possibility of SAP18 indirectly binding to A671 through a larger protein complex. This concern may be relieved by simply repeating the pulldown assay using recombinantly produced, purified SAP18 or using this recombinant SAP18 to do another direct binding assay such as isothermal titration calorimetry, biolayer interferometry, etc. As such, the language in the manuscript should be altered to reflect this lower level of certainty or the experiment should be performed.

We agree, and thank the reviewer for this suggestion. As requested, we have performed pull-down experiments using recombinant SAP18, and demonstrated that similar to what we previously showed with cell extracts, A671 directly binds SAP18 (see new Fig. 5b and explanation in page 10).

2) Based on the provided data, I do not believe there is sufficient evidence to claim that SAP18 directly regulates levels of SIRT3 in response to A671. The authors show a decrease in SIRT3 expression in response to A671, as well as a concomitant increase in SAP18 expression. Additionally, the authors show ChIP-seq. data revealing an enrichment of SIN3 at the SIRT3 promoter. While these events are certainly correlated, it is improper to claim that SIRT3 is directly regulated by SAP18 in response to A671 without a ChIP-seq. experiment treating cells with A671 and then showing an increase of SAP18 at the SIRT3 promoter relative to untreated cells. As such, the language in the manuscript should be altered to reflect this lower level of certainty or the experiment should be performed.

We agree that despite strong evidence showing regulation of SIRT3 by SAP18 at the promoter level, we cannot categorically rule out other or different mechanisms

of regulation. Therefore, as the reviewer suggested, we toned down the language in the text to reflect this uncertainty and suggest a partial, not complete, role for SIRT3 in drug-mediated tumor inhibition (Page 13).

In line with the above observation, we also examined whether the SAP18/SIRT3/SIN3 genes are critical for survival using DepMap CIRSPER database of cell lines. We found that while SAP18 and SIN3 are critical for survival, SIRT3 is not (new supplementary Fig. 6). We then knockdown SIRT3 in cells and showed that while viability was not compromised, proliferation was significantly inhibited (new supplemental Fig. 7). These new results suggest that SIRT3 expression is necessary for proliferation, but growth suppression by A671 is partially mediated through this sirtuin (see page 8 and page 13). ChIP-seq analysis is complicated by the complexity of the SIRT3 promoter with multiple transcription initiation sites. Identification of other targets of SAP18/SIN3 is also a priority for future investigation (see page 13).

3) that the authors need to tone down the language used when discussing the molecular mechanisms of A671 action. Specifically, while it is appropriate to say that A671 is a novel anticancer compound, I do not believe it is appropriate to draw conclusions about specific molecular targets without further experimental evidence other than to say that A671 is an inhibitor of class III HDACs. Moreover, while there are certainly changes in the expression of multiple HDACs in response to A671, it is not appropriate without further experimental evidence to suggest a specific protein (SAP18) is directly responsible for this. If the authors wish to keep the text of the manuscript as it is now, they should provide more direct evidence of their proposed biochemical mechanisms.

Our results strongly demonstrate that the SAP18/SIN3/SIRT3 axis plays a critical role in drug-induced growth inhibition. However, we cannot exclude the possibility that other genes regulated by the SAP18/SIN3 complex, whose expression may be altered by the drug, also cooperate with SIRT3 loss to induce cell death. As this reviewer suggested, we tuned down the language in the revised manuscript to reflect this view (see discussion, page 13 and results page 8).

In addition to T-cell lymphoma and erythroleukemia, bioinformatic analysis has identified negative correlation between SAP18 and SIRT3 in B-Cell Acute Lymphoblastic Lymphoma/Leukemia (B-CALL). We now provided new evidence that indeed a cell line, OCI_LY10, derived from this type of lymphoma and Raji (a B-cell lymphoblastoid cell line) exhibits strong sensitivity to A671 (See Fig. 1b and explanation in page 12). These new data further extend the potential usage of this compound for the treatment of leukemia and lymphoma.

Reviewer #2 (Remarks to the Author):

Summary

The authors have shown a novel anticancer therapeutic, A671, which appears to stabilize the SIN3 complex by binding SAP18. The authors show one transcriptional consequence of this mechanism is the suppression of SIRT3, and attribute the anticancer activity of A671

to this reduction in SIRT3 abundance.

Overall comments

The manuscript is well written, and the impact on the overall cancer burden in mice is especially compelling that this molecular strategy could be a useful addition to our armament of anticancer drugs. The mechanism is interesting, but is so far largely based on biased (literature informed) experiments, and lacks any genome-wide data that would give a more compelling and complete view of the molecule's mechanism. Many experiments are very well done – I especially love the rescue experiment in Fig 4e showing shRNA of SAP18 attenuates the activity of A671, a critical and clear experiment that argues a strong link between their proposed mechanism and the anticancer effect of the drug. Overall, I support publication once a few key concerns are addressed.

Major points

1. SIN3, a complex containing HDAC1/2, is referred to here only as a repressive complex. However, important new insights in this field are shifting this view. For instance, HDAC1/2 (and HDAC3) are all bound in the epigenome at the strongest enhancers and at the promoters of the most actively transcribed genes (2009, Cell, Wang, Z. et al. Genome-wide Mapping of HATs and HDACs Reveals Distinct Functions in Active and Inactive Genes and 2019, Nature Genetics, Gryder, B. E. et al. Histone hyperacetylation disrupts core gene regulatory architecture in rhabdomyosarcoma). The authors need to provide some evidence rule out (or, rule in) the possibility that their drug A671 isn't disrupting a potential need of SIN3 for maintenance of the most active and most lineage specific genes. RNA-seq analysis +/- A671, followed by GSEA analysis showing gene sets that go up as well as those that go down would be illuminating here. Both general gene sets, as well as those specific for EL4 cells (<https://www.ncbi.nlm.nih.gov/geo/query/acc.cgi?acc=GSE66343>), could be analyzed. Direct comparison for Entinostat and Nicotinamide RNA-seq in EL4 cells would be even more exciting and informative for mechanism. It is possible that SAP18/SIN3 has either activating or repressive roles depending on genomic context, and this consideration needs to be included. If the authors believe that the SIN3/SAP18 complex is more repressive than the other HDAC1/2 containing complexes (such as NuRD) that have been shown to be required for active gene expression (read and take into consideration the references above), then that too would be a helpful conclusion that RNA-seq data could offer.

We thank the reviewer for providing this insightful information and references. Indeed, while the involvement of SIN3 complex in regulation of active or lineage specific genes is an important question, our data show that expression of many redundant and active genes such as ERK, AKT, MYC and GAPDH is not affected or lightly altered by A671 (Figure 2a). Expression of HDAC1 and HDAC2, both components of the SIN3 complexes, is also not significantly affected after drug treatment (See below). The SIN3 complex may therefore be involved in specific gene regulation, an important question that would require major investigation in future studies. Indeed, a recent paper suggests a role for SIN3 in specific gene regulation during hypoxia (Tiana M, Acosta-Iborra B, Puente-Santamaría L, et al. The SIN3A

histone deacetylase complex is required for a complete transcriptional response to hypoxia. *Nucleic Acids Res.* 2018;46(1):120-133).

Our model specifically proposes that the suppression effect of the SIN3 complex is elevated through direct binding of A671 to SAP18 (see model figure 6a). Higher repression of SIN3 by A671 likely affect the expression of many genes associated with cell survival. Our results warrant further analysis to determine the effect of SAP18 on global gene expression (see discussion, page 13). Our results and suggested model uncover a major mechanism by which SAP18 in the context of SIN3 or non-SIN3 complexes regulates gene expression.

2. Furthermore, this could hint at a more complete explanation of the way the drug downregulates Sirt3 while upregulating Sirt1/2 (which is itself very interesting). The relative expression is nice, but what is the absolute expression of Sirt1/2/3? If Sirt1/2 start at a low expression, and Sirt3 starts at a much higher level, then that can be thought of differently then if all three genes start at the same levels yet respond so differently. Again, getting transcripts per million from RNA-seq of the Sirt genes +/- A671 could be helpful by giving a sense of the absolute, not just relative, mRNA levels.

Indeed as noted by the reviewer, upregulation of SIRT1/2 is an interesting observation we plan to investigate in future studies. While we used both HEL and EL4 for q-RT-PCR analysis (Fig. 3), our RNAseq data was originally performed using HEL cells that are highly sensitivity to the compound. We used this cell line because it represents human leukemia with available expression databases. In RNAseq analysis, the level of SIRT1 and SIRT2 mRNA in leukemic cells is moderate and relatively similar to SIRT3. This observation is further supported by mRNA expression level in HEL cells from the human protein Atlas (www.proteinatlas.org). It would be interesting to determine if induction of SIRT1/2 by A671 contributes to tumor regression by knocking down these genes back to untreated levels. Indeed, there is evidence suggesting that SIRT2 may act as a tumor suppressor (Kim HS, Vassilopoulos A, Wang RH, et al. SIRT2 maintains genome integrity and suppresses tumorigenesis through regulating APC/C activity. *Cancer Cell.* 2011;20(4):487-499). SIRT1 is also known to act as an oncogene or tumor suppressor, depending on cellular context (Fang Y, Nicholl MB. Sirtuin 1 in malignant transformation: friend or foe?. *Cancer Lett.* 2011;306(1):10-14). Future studies using our novel compound should provide important insight into the role of these sirtuins in cancer progression.

3. A central claim is that SIRT3 downregulation is responsible for the anticancer impact of A671. While there are experiments knocking down SAP18, I didn't see experiments where they knocked out SIRT3 and showed reduction in cell growth. I apologize if I missed this, but it would be essential to making the central claims of the paper.

As this reviewer suggested, we knocked-down SIRT3 and showed significant suppression of cell proliferation (Supplementary Fig. 7 and explanation in page 8). Since SIRT3 expression is downregulated by A671, this experiment further supports a role for SIRT3 in drug-induced inhibition of cell proliferation.

4. The authors claim A671 is a HDAC-class III inhibitor. But this isn't true: it is a molecular glue for SAT8/SIN3. It is a HDAC-class III transcriptional suppressor, not an inhibitor.

We agree with this comment and have changed the text accordingly in pages 6 and 7).

5. A central claim is the "leading to recruitment of the SAP18-SIN3 complex to the SIRT3 promoter and its transcriptional silencing". Yet, this notable claim lacks any supporting data as far as I can see. To make this novel claim they need ChIP-seq or at least ChIP-qPCR for SAP18/SIN3 at the SIRT3 promoter (plus primers for a negative control region and primers for the SIRT1/2 promoters), +/- A671.

We agree with this critique and while our data support regulation of SIRT3 by SAP18-SIN3, we still have to prove direct binding of this complex to the SIRT3 promoter. This is a challenging experiment considering the fact that SIRT3 uses different transcription start sites. We tuned down the language in the manuscript by suggesting that SIRT3 transcription is in part suppressed by drug-induced activation of the SAP18/SIN3 effect. The fact that SIRT3 knockdown significantly inhibited cell proliferation (new supplementary Fig. 7) further supports the notion that SIRT3 is at least in part responsible for the growth inhibitory effect of the compound. We discussed this issue on page 13.

Novelty/notability claims

6. The papers novelty is high and the authors have followed a mostly well informed and courageous line of experimentation. I cannot find prior work that detracts from the overall novelty of this study.

Thank you.

7. The pulldown experiments are excellent additions that make the novel claims of direct binding to SAP18/SIN3 well supported.

To address this insightful comment, we performed pull down experiments using recombinant SAP18 and the results were very similar to pull down using

extract. Thus, A671 directly bind SAP18 independently of other proteins (new figure 5b and page 10).

Minor points

8. The interpretation of Fig 4f is oversold quite a bit. More careful analysis of actual ChIP-seq data would be needed to draw such a strongly worded conclusion of gene specificity. Not to mention – promoter binding accounts for only a small fraction of functional binding events in the genome! Differences in SIN3/SAP18 binding in enhancers could be even more important for gene-selective effects of the complex.

We agree and as suggested, we tuned down this statement on page 13.

9. The authors make statements about the importance of Sirt3 to support tumor growth, and claim SAP18 at high levels improves outcomes by keeping SIRT3 low in human cancers. Can the authors make use of the DepMap CRISPR data for SIRT3/SAP18/SIN3 across the hundreds of cancer cell lines in that publicly accessible.

We thank the reviewer for this suggestion; we searched this database for SAP18 and SIRT3 function for cell survival. While *SAP18* and *SIN3* were found critical for survival, SIRT3 was not (new supplemental Fig. 6). Accordingly, we now show that knock-down of SIRT3 significantly inhibited cell proliferation (new supplementary Fig. 7), but not survival. This result suggests a partial role for SIRT3 in drug-induced suppression of cell proliferation. We have changed the text to reflect these new observations (page 8 and page 13).

In addition to T-cell lymphoma and erythroleukemia, bioinformatic analysis has identified negative correlation between SAP18 and SIRT3 in B-Cell Acute Lymphoblastic Lymphoma/Leukemia (B-CALL; Fig. 6). We now provide new evidence that OCI_LY10, a cell line derived from this type of lymphoma, and Raji (a B-cell lymphoblastoid cell line) exhibit strong sensitivity to A671 (See Fig. 1b and explanation in page 12). These new data further extend the potential usage of this compound for the treatment of leukemia and lymphoma.

Figure comments

10. Figure 1a shows 3 molecules, but Fig 1b shows data from only 1 molecule. Help the reader by indicating in Fig 1b which molecule from Fig 1a is being tested (in the figure itself, not just in the legend). Same for Fig 1c. Fig 1d-1f all have A671 labeled so no need to change those.

We corrected this mistake in all figures to show which drug is used for each particular experiment.

11. Fig 1c please indicate cell line used in the figure itself, not just the legend. Same for

Fig 1d-f.

We added the name of cell line in each figure, as requested.

12. Fig 2b-c have the same axis labels and the same conditions: please also indicate the tumor model so the difference between these two is apparent.

We corrected this figure as suggested.

13. Figure 3a etc.: please indicate on the figure itself that this is an HDAC activity assay. Just add “HDAC type X activity” to the Y-axis label, above “Luminescence”. Or, group a and b, and c and d, and have a hovering title for HDAC type activity over each pair.

We fixed these figures as suggested.

14. In Figure 3 and others, many panels say “level”. Can you additionally indicate in the figure axis labels either “mRNA level” or “protein level”, whichever it is?

We corrected this concern in figure 3 and other relevant figures uniformly.

15. The model image needs a lot of work. Power point was used, but Adobe Illustrator would give the authors a much cleaning and eye-catching image. It’s worth the investment, in my opinion.

We thank the reviewer for this suggestion. We believe the revised new model (Fig. 6a) is much better now.

REVIEWERS' COMMENTS:

Reviewer #1 (Remarks to the Author):

The authors were able to adequately address my concerns expressed in the previous communication. In light of the updated information provided by the authors, I believe the new manuscript is suitable for publication.

Reviewer #2 (Remarks to the Author):

Well done completing the rebuttal, I hope the process was helpful to your scientific insight. The paper is much improved in my opinion.

30/09/2020

REF: COMMSBIO-20-1030B, A C₂₁-steroidal derivative suppresses T-cell lymphoma in mice by inhibiting SIRT3 via SAP18-SIN3

Reviewers' comments:

Reviewer #1 (Remarks to the Author):

The authors were able to adequately address my concerns expressed in the previous communication. In light of the updated information provided by the authors, I believe the new manuscript is suitable for publication.

We thank the reviewer for understanding our concerns and accepting the work for publication in the journal of “*Communications Biology-Nature*”. Your productive criticisms are the energy booster for our future endeavors, as well.

Reviewer #2 (Remarks to the Author):

Well done completing the rebuttal, I hope the process was helpful to your scientific insight. The paper is much improved in my opinion.

We express our gratitude for the assistance that you provided to improve our manuscript to the current standard.